# Impact of *IsaA* Gene Disruption: Decreasing Staphylococcal Biofilm and Alteration of Transcriptomic and Proteomic Profiles

**DOI:** 10.3390/microorganisms10061119

**Published:** 2022-05-29

**Authors:** Pei Yee Ma, Chun Wie Chong, Leslie Thian Lung Than, Anita Binti Sulong, Ket Li Ho, Vasantha Kumari Neela, Zamberi Sekawi, Yun Khoon Liew

**Affiliations:** 1School of Postgraduate Studies, International Medical University, Kuala Lumpur 57000, Malaysia; ma.peiyee@student.imu.edu.my; 2School of Pharmacy, Monash University Malaysia, Subang Jaya 47500, Malaysia; chong.chunwie@monash.edu; 3Department of Medical Microbiology, Faculty of Medicine and Health Sciences, University Putra Malaysia, Serdang 43400, Malaysia; leslie@upm.edu.my (L.T.L.T.); vasantha@upm.edu.my (V.K.N.); zamberi@upm.edu.my (Z.S.); 4Department of Medical Microbiology and Immunology, Pusat Perubatan UKM, Kuala Lumpur 56000, Malaysia; dranita@ppukm.ukm.edu.my; 5Department of Life Sciences, School of Pharmacy, International Medical University, Kuala Lumpur 57000, Malaysia; hoketli@imu.edu.my

**Keywords:** *Staphylococcus aureus*, IsaA, gene disruption, phenotype, proteomic, transcriptomic

## Abstract

*Staphylococcus aureus* expresses diverse proteins at different stages of growth. The immunodominant staphylococcal antigen A (IsaA) is one of the proteins that is constitutively produced by *S. aureus* during colonisation and infection. SACOL2584 (or *isaA*) is the gene that encodes this protein. It has been suggested that IsaA can hydrolyse cell walls, and there is still need to study *isaA* gene disruption to analyse its impact on staphylococcal phenotypes and on alteration to its transcription and protein profiles. In the present study, the growth curve in RPMI medium (which mimics human plasma), autolytic activity, cell wall morphology, fibronectin and fibrinogen adhesion and biofilm formation of *S. aureus* SH1000 (wildtype) was compared to that of *S. aureus* MS001 (*isaA* mutant). RNA sequencing and liquid chromatography–tandem mass spectrometry were carried out on samples of both *S. aureus* strains taken during the exponential growth phase, followed by bioinformatics analysis. Disruption of *isaA* had no obvious effect on the growth curve and autolysis ability or thickness of cell walls, but this study revealed significant strength of fibronectin adherence in *S. aureus* MS001. In particular, the *isaA* mutant formed less biofilm than *S. aureus* SH1000. In addition, proteomics and transcriptomics showed that the adhesin/biofilm-related genes and hemolysin genes, such as *sasF*, *sarX* and *hlgC*, were consistently downregulated with *isaA* gene disruption. The majority of the upregulated genes or proteins in *S. aureus* MS001 were *pur* genes. Taken together, this study provides insight into how *isaA* disruption changes the expression of other genes and has implications regarding biofilm formation and biological processes.

## 1. Introduction

Humans serve as a major reservoir for *Staphylococcus aureus*. This important opportunistic pathogen is a member of normal human flora and is generally found on the skin, mucous membrane and especially in the anterior nares of 20–40% of healthy adults [1,2]. Reports have shown that approximately 20% of humans are persistent *S. aureus* carriers without knowing it, as asymptomatic colonisation is very common. In addition, strains of *S. aureus*, some more virulent than others, can be detected from various sources [3,4,5]. During hospitalisation, they can attach to or persist on host tissues or implanted materials. Healthcare workers, immunocompromised patients, intravenous drug users and hospitalised patients (particularly those with long-term indwelling intravascular catheters) tend to have higher rates of *S. aureus* colonisation, up to 80%, as *S. aureus* typically spread by direct contact or fomites [6,7,8,9]. This indicates that they are versatile and opportunistic in nature, which often leads them to cause community-acquired and nosocomial infections, from minor skin infections to life-threatening infections [1,6,10,11]. Despite using various effective antibiotics in treatments, instances of isolating antimicrobial-resistant strains are reported periodically [6]. As a result, the treatment and management of *S. aureus* infections remains challenging as the species has the potential to evolve from single-drug resistant strains to multi-drug resistant strains (e.g., methicillin-resistant *S. aureus* (MRSA), vancomycin-resistant *S. aureus* (VRSA)) [12,13]. Moreover, their ability to adopt a diverse range of niches through the expression of protective functional proteins is considered one of the challenges in eliminating them.

Given their advantage of expressing survival proteins, researchers have extensively used ‘omic’ approaches to study several strains of *S. aureus*. It has been shown that *S. aureus* expresses a wide variety of proteins, and some of them might have a role in virulence, such as the immunodominant staphylococcal antigen A (IsaA). In early 2000, IsaA was shown to be either displayed on the cell surface or secreted into the host [14]. Ziebandt et al. revealed that most strains of *S. aureus* could produce IsaA protein. It is expressed only at low staphylococcal cell densities, which indicates that IsaA protein is expressed during the exponential growth phase [15,16]. These surface proteins are able to promote the adherence of *S. aureus* to damaged host tissue and to help evade antibody-mediated immune responses by binding proteins in the blood [10,17,18]. Further, IsaA protein is also involved in cell wall turnover [19,20]. Additionally, it is believed that the protein is associated with cell growth, separation, and survival due to the highly abundant IsaA found in the septal region of dividing staphylococcal cells [21].

On the other hand, several studies have shown that IsaA protein is against IgGs in human body serum [22,23,24,25]. Hence, IsaA protein is expressed as an antigenic molecule on the surface of staphylococcal cells, and thus it is recognisable by the IgGs of patients [22]. Another study identified that the serum IgG levels against IsaA in healthy *S. aureus* carriers were significantly higher than in those who are non-carriers [23]. Hence, IsaA is considered an immunogen that is suspected to be responsible for protecting us from opportunistic infection. In a study using an animal model, anti-IsaA monoclonal antibody (IgG1) effectively prevented murine staphylococcal infection, including in a central venous catheter-related infection model, as it can recognise all tested *S. aureus* strains including MRSA [24,25,26,27,28]. In addition to this finding, Sanne et al. also investigated that human anti-IsaA IgG1 monoclonal antibodies (mAb) can improve the survival of mice with *S. aureus* bacteraemia [29].

All these studies provide pieces of evidence pointing to the high immunogenicity of IsaA, its invariantly on the *S. aureus* cell surface and its involvement in virulence. Therefore, the present study aims to further investigate IsaA by focusing on its mutant and its impact on the growth, adhesion to human fibrinogen and fibronectin, biofilm formation, autolysis and cell morphology. The *isaA* mutant and its parent were also processed by RNA-seq and tandem mass spectrometry to investigate differential protein and gene expression.

## 2. Materials and Methods

### 2.1. Bacterial Strain and Culture Conditions

*S. aureus* wildtype (SH1000) and Δ*isaA* mutant (MS001) were obtained from Prof. Foster (The Krebs Institute, Department of Molecular Biology and Biotechnology, University of Sheffield). SH1000 was inoculated in tryptic soy broth (TSB), while MS001 was grown in TSB supplemented with 5 µg/mL tetracycline at 37 °C overnight [19]. The bacteria culture solution was supplemented with 10% glycerol and stored at −80 °C as stock.

### 2.2. Growth Curves

Overnight cultures of both strains in TSB were centrifuged, and the pellet was suspended with Dulbecco’s-phosphate-buffered saline (D-PBS) and standardised by adjusting optical density at 600 nm (OD_600_) to 1 with RPMI-1640. The suspension was serially diluted to 1:1000 with RPMI-1640 and then incubated at 37 °C with shaking in an incubator shaker at 200 rpm. Bacterial growth was measured every 0.5 h for a total of 24 h using a microplate reader. Absorbance versus time of each strain was plotted. Data analysis was performed from three independent experiments, and Baranyi modelling was used to determine the maximum bacteria density and maximum specific growth rate [30]. Statistical significance of MS001 versus parent strain (SH1000) growth at all time points was analysed by a Student’s *t*-test.

### 2.3. Autolysis Assay

Both strains were cultured in TSB overnight and were centrifuged. The pellet was resuspended in 1 M NaCl TSB and adjusted to an OD_600_ of 0.08. The cells were harvested in an incubator shaker (200 rpm) at 37 °C to OD_600_ of 0.8~1.0 and were centrifuged at 6500× *g* for 10 min. The pellet was resuspended in sterile-ice-cold milli-Q water and adjusted with 50 mM Tris–HCl buffer (pH 7.2, containing 0.05% Triton X-100) to an OD600 of 1. The cell suspension was incubated at 37 °C with shaking at 125 rpm. At 30 min intervals, the progression of lysis was measured by using a spectrophotometer at OD_600_, and data were reported as percentage of initial OD_600_ for each sample. Statistical significance was analysed using IBM SPSS v.28.0.0 (USA) via Student’s *t*-test at all time points for the autolytic activity.

### 2.4. Biofilm Formation

Overnight cultures grown in TSB were centrifuged, and the pellet was washed with PBS. The cells were adjusted to OD_600_ in TSB supplemented with 0.2% glucose and diluted to 1:50. A total of 200 µL of the diluted culture was loaded into each well of a 96-well flat-bottomed tissue culture plate and incubated at 37 °C for 24 h. The wells were gently rinsed three times with D-PBS, incubated at 60 °C for 15 min and stained with 0.1% crystal violet for 10 min at room temperature. The plate was rinsed three times under running tap water and air-dried. Crystal violet was solubilised with 40 mM HCl in ethanol and incubated at room temperature for 5 min. The content in the wells was mixed by pipetting, and absorbance was measured at OD_595_ with a microplate spectrophotometer. An independent triplicate experiment was performed, each being repeated at least three times, and Student’s *t*-test was done to show the differences between wildtype and mutant strains, with a *p*-value ≤ 0.05 considered significant.

### 2.5. Adhesion Assay

Human fibrinogen (10 mg/mL) and fibronectin (1 mg/mL) in D-PBS was diluted to 2.5 µg/mL. The 96-well flat-bottom enzyme-linked immunosorbent assay (ELISA) plates coated with fibrinogen and fibronectin were incubated at 4 °C overnight. The coated wells were washed with PBS-0.05% Tween 20 (PBS-T) and were blocked with PBS-T supplemented with 1% bovine serum albumin at room temperature for one hour. The wells were rewashed with PBS-T. Bacterial cultures in TSB were grown until an OD_600_ of 1, centrifuged, and washed with PBS-T. The pellet was resuspended to an OD_600_ of 0.45 in TSB-0.4% glucose. Adjusted cultures were pipetted into the coated plate and incubated at 37 °C for one hour. Wells were washed aggressively with PBS-T and BacTiter-Glo^TM^ Reagent, which generated a glow-type luminescent signal from the BacTiter Glo kit added to each well. The content was mixed with an orbital shaker and incubated for 5 min. A luminescence microplate reader was used to measure the amount of bacterial ATP according to the manufacturer’s guidelines in order to quantify the number of bacteria adhered to the plate-bound fibrinogen or fibronectin. This experiment was carried out at least in triplicate. The Student’s *t*-test was used to compare means of bacterial ATP.

### 2.6. Transmission Electron Microscopy (TEM) Analysis

Bacteria cultures in TSB were grown to the log phase at 37 °C and centrifuged at 3900× *g* for 5 min. The pellet was resuspended in 1 mL of 3% (*v*/*v*) glutaraldehyde in 0.1 M sodium cacodylate. Glutaraldehyde-fixed samples were stored at 4 °C for 4 days before being washed twice with 0.1 M sodium cacodylate. After washing, the samples were postfixed with 1.0% osmium tetroxide containing 1.5% potassium-ferrocyanide in 0.1 M cacodylate buffer for 2 h at 4 °C. Samples were rinsed in distilled water, treated with 1% aqueous uranyl acetate (*w*/*v*) in 50% ethanol for one hour at 4 °C and dehydrated in ascending series of graded ethanol dilutions to 100%. Then, dehydrated samples were embedded into Spurr resin and left to polymerise in an oven at 60 °C for 24 h. Resins were sectioned by cutting 30 nm films at 25 °C using an ultramicrotome. Subsequently, the ultrathin sections were mounted on carbon-coated copper grids and observed with a JEOL JEM-2100F transmission electron microscope. Cell wall thickness was measured from photographic images at a final magnification of at least ×20,000.

### 2.7. RNA Preparation and DEG Analysis

#### 2.7.1. Total RNA Extraction

Bacteria was grown in TSB at 37 °C and harvested during the exponential growth phase, then centrifuged at 3900× *g* for 10 min at 4 °C. The pellet was resuspended with Tris–EDTA (TE) buffer containing 2 mg/mL Lysostaphin and 50 mg/mL lysozyme and incubated at 37 °C for one hour. An equal volume of RNA lysis buffer was added to the incubated mixture and briefly vortexed. The mixture was centrifuged at 16,000× *g* for 2 min, and proceeded to the RNA binding and elution step as instructed in the Monarch Total RNA Miniprep Kit. The amount of total RNA was determined by NanoVue Plus (GE Healthcare), followed by checking the quality of the extracted RNA using the Bioanalyzer Nano RNA chip with an RNA integrity number (RIN) > 8.0 for cDNA library preparation prior to RNA-sequencing (RNA-seq).

#### 2.7.2. Differentially Expressed Gene (DEG) Analysis

The quality of sequencing raw data were inspected using FastQC (version 0.11.8, Cambridge, UK). Low-quality reads (with Phred score less than Q20) and adapter and poly G sequences were removed using Fastp [31]. The clean reads were aligned against a *S. aureus* reference genome from the NCBI database using HISAT2 [32]. The reference genome index was built using the accessory script of the HISAT2 package. The aligned RNA-seq reads in BAM format were quantified using featureCounts [33]. DESeq2 was adopted to perform differentially expressed gene (DEG) analysis. To compare expression levels between samples, raw read normalisation was performed. DESeq2 [34] performed an internal normalisation where the geometric mean was calculated for each gene across all samples. Before running DEG analysis, sample-level and gene-level QC were performed on the count data. At sample-level QC, principal component analysis (PCA) was constructed to identify any potential outliers. At gene-level QC, genes with read counts of zero or with a mean less than ten were omitted from downstream analysis. DESeq2 fits negative binomial generalised linear models for each gene and uses the Wald test for significance testing. DEGs with *p*-adjusted value ≤ 0.05 and log_2_ fold change ≥1 and ≤−1 were categorised as significant. The significant DEGs were selected to perform Gene Ontology and Pathway Analysis using EnrichR or clusterProfiler [35]. The hierarchical heatmap was plotted to represent the expression of the significant DEGs in all the samples.

### 2.8. Quantitative Proteomic Analysis

#### Protein Preparation, Digesting and LC–MS/MS Analysis

Bacteria grown in TSB at 37 °C was harvested at the log-growth phase and centrifuged at 3900× *g* for 15 min. The pellet was washed with detergent extraction buffer containing 25 mM Tris-HCl (pH7.2), 10 mM CHAPS, 0.5 M NaCl, 5% glycerol and 1 mM phenylmethylsulfonyl fluoride (PMSF) and lysed with homogeniser through 0.1 mm glass beads. Cell debris was removed by centrifuging the mixture in the tubes at 15,000× *g* for 15 min at 4 °C. The supernatant was transferred and mixed well with six volumes of ice-cold acetone with trichloroacetic acid. The pellet was dried. Protein quantitation of each sample was performed using a BCA assay.

Sample digestion followed the S-Trap Mini Spin digestion protocol outlined by the manufacturer (ProtiFi, New York, NY, USA). The obtained peptides were dried by vacuum centrifugation and then reconstituted in 200 mM HEPES pH 8.8. Peptide concentration was determined using the Pierce quantitative colourimetric peptide assay (Thermo Scientific, Waltham, MA, USA). Tandem mass tag (TMT) reagent labelling of the samples was performed as per the manufacturer’s (Thermo Scientific, Waltham, MA, USA) instructions. Samples were reconstituted in 2% ACN 0.1% formic acid and analysed by LC–MS/MS (Q-Exactive HFX, Thermo Fisher, Waltham, MA, USA). A normalisation factor was obtained from the label check experiment, and the original TMT-labelled peptide samples were then pooled at a 1:1 ratio across all individual samples.

The combined sample mixture was vacuum dried and resuspended in 0.1% trifluoroacetic acid then fractionated by Pierce High pH Reversed-Phase Peptide Fractionation Kit (Thermo Scientific, Waltham, MA, USA) according to the manufacturer’s instructions. Each fraction was vacuum centrifuged to complete dryness and then reconstituted in 0.1% formic acid for LC–MS/MS.

LC–MS/MS was performed by Q-Exactive HFX, Thermo Fisher, Waltham, MA, USA. The peptide mixture in each fraction was injected onto the peptide trap and washed with loading buffer for 10 min. The peptide trap was then switched in line with the analytical nano-LC column. Peptides were eluted from the trap onto the nano-LC column, and peptides were separated with a linear gradient of 5% mobile phase B to 30% mobile over 110 min at a flow rate of 600 nL/min, followed by 85% B for 8 min. The column was re-equilibrated using 5% mobile phase B.

The column eluent was directed into the ionisation source of the mass spectrometer operating in positive ion mode. Peptide precursors from 350 to 1850 *m*/*z* were scanned at 60 k resolution. The 10 most intense ions in the survey scan were fragmented by higher energy collisional dissociation (HCD) using a normalised collision energy of 33 with a precursor isolation width of 0.8 *m*/*z*. Only precursors with charge states +2 to +5 were subjected to MS/MS analysis. MS/MS scan resolution was set at 45 k and dynamic exclusion was set to 30 s.

The MS/MS data files were processed using Proteome Discoverer (Version 2.1.0.81, Thermo Scientific, Waltham, MA, USA). The data were searched using search engine SequestHT against *S. aureus* (bacteria) proteins from the Uniprot database. All data used for protein identification were evaluated by the peptide-to-spectrum match false discovery rate (PSM FDR) ≤ 1% analysis. A pairwise-relative abundance comparison was performed. Statistical significance of protein was measured using the Student’s *t*-test, and differences were considered significant at *p*-value ≤ 0.05 and fold change greater than ±1.2, at which point they were recognised as differentially expressed proteins.

## 3. Results

### 3.1. Growth Pattern and Autolysis of S. aureus isaA Mutant

There were no significant changes in the growth and autolysis curves upon *isaA* gene disruption (Figure 1). However, the Δ*isaA* mutant showed a slight reduction in maximum bacterial density as determined by Baranyi modelling, such as OD_600_ values of 1.24 and 1.46 for mutant and wildtype, respectively. The maximum specific growth rate of both *S. aureus* strains was the same (≈0.5 h^−1^), but the *isaA* mutant had a slightly prolonged lag phase, as illustrated in Figure 1a. Similarly, *isaA* gene disruption only reduced the autolysis rate to a certain extent for *S. aureus* relative to the wildtype (Figure 1b). For example, the Δ*isaA* mutant exhibited 75% autolytic activity, while its parental strain showed about 84% autolysis in 60 min of Triton X-100-induced lysis at 37 °C, but the results were not significance (*p* = 0.096). However, the phenotype with reduced autolysis (Δ*isaA* mutant) had slight growth retardation with a more prolonged lag phase and lower final optical density.

### 3.2. Cell Wall Morphology of Wildtype and Mutant Strains

The Δ*isaA* mutant cells maintained a cocci shape with a smooth cell surface, similar to wildtype cells. Microscopic inspection also showed no changes to cell morphology in the absence of the *isaA* gene (Figure 2). Additionally, the thickness of the *S. aureus* cell wall was unaffected with *isaA* gene disruption (32.03 ± 1.63 nm and 31.38 ± 1.22 nm for *S. aureus* Δ*isaA* mutant and wildtype, respectively).

### 3.3. Adherence of Wildtype and Mutant Strains to Fibronectin and Fibrinogen

In the *isaA* mutant, staphylococci tend to adhere to the tested fibronectin or fibrinogen, where overall ATP bioluminescence was much higher than that of the wildtype *S. aureus*. Importantly, there was a statistically significant difference between the binding of Δ*isaA* mutant and its wildtype strain on fibronectin. Figure 3 illustrate representative amounts of released bacterial ATP detected in different extracellular matrix components which clearly shows a different ATP profile.

### 3.4. Capability of Biofilm Formation

The *isaA* gene strongly influenced biofilm formation; the Δ*isaA* mutant is less likely to produce biofilms than its parental strain SH1000 (Figure 4). Generally, there was an approximately 70% reduction in biofilm formation in the *isaA* mutant when grown in TSB medium supplemented with 0.2% glucose. Significant biofilm was only observed for *S. aureus* SH1000 and not the Δ*isaA* mutant.

### 3.5. Differentially Expressed Protein and Gene Analysis

Volcano plots of fold changes were created to view variations in gene and protein expression between *isaA* mutant and wildtype *S. aureus* (Figure 5a,b). A total of 135 DEPs were found in the *S. aureus* proteome using tandem mass spectrometry (LC–MS/MS), where 55 were downregulated proteins and 80 were upregulated proteins in the Δ*isaA* mutant. Apart from the proteome, high-throughput RNA sequencing (RNA-seq) identified 514 DEGs when the *isaA* gene was disrupted, of which 285 were downregulated and 229 were upregulated. For proteomics, several proteins were more downregulated in the Δ*isaA* mutant than in the wildtype strain, such as those involved in immunoglobulin binding (protein A (SpA), Staphylococcal binding immunoglobulin protein (Sbi)) and adherence to host cells (serine–aspartate repeat containing protein D (SdrD)), as well as lipid modifying (lipase 2, SAL2). In addition, transcriptomics found that mRNA corresponding to proteases, staphylococcal hemolysins and adhesive surface protein were highly downregulated in the Δ*isaA* mutant, at 19.7, 16.9, 15.8, 14.5, 12.3, 12.1, 11.9, 9.4, 9.4 and 9.1-fold for *splE*, *splD*, *splC*, *splB*, *splA*, *hlgB*, *hld*, *hlgC*, *splF* and *clfA* genes, respectively.

GO enrichment analyses unravelled nine significant GO terms, which were classified into the category of biological process (BP). These included the top three enriched terms of “multi-organism cellular process (GO:0044764)”, “cell killing (GO:0001906)” and “cytolysis (GO:0019835)”. On the other hand, there is only one significant GO term enriched into the category of cellular component (CC) with 11 DEGs. As shown in Figure 5c, this particular GO term is “extracellular region (GO:0005576)” involving the genes associated with virulence factors, such as hemolysins and superantigen-like proteins. All these significant GO terms in BP and CC were enriched from the downregulated genes. The PCA of both omics showed that samples could be generally clustered into two groups, either wildtype or mutant (Figure 5d). The dots labelled as RNA_W or Prt_W are the dataset of wildtype *S. aureus* generated from transcriptomics and proteomics, respectively, while RNA_M or Prt_M are the dataset from the Δ*isaA* mutant. All these datasets were obtained from three replicates. It was noted that the RNA-seq data and proteomics were further grouped into representative small groups within each cluster (Figure 5d). This indicates high levels of reproducibility between replicates within each omics approach. Moreover, the correlation between mRNA and protein levels was moderately positive, with correlation coefficient of *r_s_* = 0.613 for the matched DEGs and DEPs (Figure 5e).

The co-expression of genes for the production of corresponding proteins is depicted as a heatmap in Figure 6. It found that not all the matched mRNA and protein for the respective gene necessarily has the same differential expression profile. For example, SAOUHSC_00114 was downregulated according to transcriptomics, but was slightly upregulated in the Δ*isaA* mutant based on proteomics. This opposite expression pattern could also be observed for other genes, as shown in Figure 6. However, a consistent expression pattern was found between the transcripts and proteins for the Δ*isaA* mutant. There were eight genes and proteins consistently downregulated, while another 18 upregulated proteins also demonstrated a pattern similar to their corresponding mRNA transcription profile.

## 4. Discussion

Understanding the associated phenotypes, protein or gene expression profiles change upon the *isaA* gene disruption is important in guiding the development of an effective way to treat or limit the spread of the *S. aureus*. Previous studies suggest that the *isaA* are expressed during infection as well as colonisation; however, the disruption of the *isaA* gene on its influence on proteome or transcriptome alteration of *S. aureus* is still unclear.

Here, we present the disruption of the *isaA* gene; there were significant differences in certain phenotypes for the *isaA* mutant strain. However, the growth pattern, autolysis profiles and cell wall morphology were not greatly influenced, but some differences were observed in this study for growth and autolysis profiles. As reported, IsaA was found to have autolytic activity that can assist in cell wall expansion, turnover, growth and cell separation. Besides, researchers found that cell wall turnover is dependent on autolytic activity [19,36]. Therefore, as shown in Figure 1a, the growth curve of the *isaA* mutant has a slightly lower (but not significant) maximum bacterial density than the wildtype strain. Similarly, autolytic activity in the *isaA* mutant was slightly decreased in the present study. The reduced autolysis might explain the growth retardation of the *isaA* mutant by slowing the cell replication, which leads to decreased bacterial density. Importantly, the downregulation of the *nrdI* gene in the Δ*isaA* mutant as determined by proteomics and RNA-seq may also be one reason for the reduction in maximum bacterial density. It is known that the *nrdI* gene product is involved in ribonucleotide reductase function, which is responsible for replication and adaptation to different phases in the growth curve [37,38,39]. In the exponential to stationary phase they even have similar maximum specific growth rates or slopes, but a prolonged lag phase for the *isaA* mutant was observed. Disruption of the *isaA* gene caused the cells to require more time to adapt gene regulation during the lag phase, thus growth retardation occurred with its modified autolytic rate for cell wall turnover and downregulated *nrdI* gene. However, no changes were observed in the cell wall thickness and morphology. Undeniably, the loss of IsaA function alone may still be able to be compensated by other gene products with similar functions for cell wall expansion, turnover, growth and cell separation. For example, transglycosylase (SceD), cell wall hydrolase (LytN), and autolysins (Atl, Sle1) might have compensatory roles in autolytic activity and are expressed for cell division in *S. aureus* [19,40,41,42]. Based on our transcriptomic data, the genes of *atl*, *sle1* and *lytN* were not affected by *isaA* gene disruption, while the *sceD* gene had increased expression.

In the present study, significant reduction in biofilm formation for the Δ*isaA* mutant under static conditions is in good agreement with previous data showing that deletion of *isaA* significantly decreased biofilm formation. A number of studies have shown that the expression of intercellular adhesion (ica), fibrinogen-binding proteins (Fib), fibronectin-binding proteins (FnBPs), collagen-binding protein (cna), Sdr proteins, clumping factors (Clfs), bone sialoprotein-binding protein (Bbp), SpA and *S. aureus* surface proteins (Sas) are associated with biofilm production [43,44,45,46]. Notably, our proteomic data showed that the production of SdrD and SpA was significantly reduced in the *isaA* mutant strain. A study carried out by Merino et al. also indicated that SpA is a protein that might affect biofilm formation in *S. aureus*. The deletion of the *spa* gene significantly reduced staphylococcal biofilm production [47]. Markedly downregulated genes coding for SasF, Fib and ClfA were also detected in the Δ*isaA* mutant in this present study. It was previously shown that ClfA plays an important role as an adhesin for the early stages of biofilm formation, and SasF was recently found to be a component of the biofilm matrix [44,48]. Although Fib was significantly reduced in the Δ*isaA* mutant, it did not decrease the binding of *isaA* mutant to fibrinogen. In contrast, *isaA* gene disruption further enhanced the binding of *S. aureus* to these tested extracellular matrix components (fibrinogen and fibronectin), irrespective of the downregulation of the *fib* gene. However, the *fnbA* gene was significantly upregulated, with approximately a 3 log_2_ fold change in the Δ*isaA* mutant. The highest binding of the *isaA* mutant to extracellular matrix components, especially fibronectin, might therefore be related to the upregulation of the *fnbA* gene. Furthermore, the other genes, such as *eap*, *coa* and *vWbp*, can compensate for the function of Fib as well [49,50,51].

Protein analysis revealed greatly reduced Sbi synthesis (along with SpA and SdrD as mentioned above) in the Δ*isaA* mutant. Sbi is the main key for staphylococcal immune evasion. Additionally, a few virulence factor genes have been shown to be highly downregulated with *isaA* gene disruption, including genes responsible for serine proteases (SplA to -F) and hemolysins (gamma- and delta-hemolysin) as found in the present transcript database. Several studies reported that these serine proteases cause many clinical infections, such as endocarditis, pneumonia, and allergic asthma [52,53,54]. Meanwhile, hemolysins can lyse mammalian cell membranes to escape killing by phagocyte and cause septic arthritis in the host [55,56,57,58,59,60]. Our enrichment analysis results indicate that a few more genes encoding virulence factors were significantly enriched in the BP or CC categories, such as virulence factor alpha-hemolysin, leukocidins, leukotoxin, superantigen-like proteins and hyaluronate lyase. The role of these virulence factors has been well documented elsewhere [61,62,63,64,65]. For example, hyaluronate lyase plays a vital role in subcutaneous infections; hemolysin, leukocidins and leukotoxin might exhibit lytic activity on host cells and subsequently evade the host immune response. The role of superantigen-like proteins and their association with *S. aureus* pathogenesis are comprehensively discussed in the literature written by Fraser and Proft [66]. Basically, superantigen-like proteins can lead to severe life-threatening conditions.

It is unsurprising that the transcriptome and proteome were moderately related rather than being strongly correlated. This partial discordance supports that post-transcriptional processes play a significant role in adapting to growth conditions and protein synthesis. It could also be because some proteins detected by proteomic analysis were not or were weakly detected in transcriptomic analysis, probably due to the mRNA’s half-life and stability. On the other hand, some insoluble proteins, low abundance proteins and post-translational factors also contribute to the inconsistent finding between protein and transcript correlation. Nevertheless, as shown in the heatmap, we still observed consistent expression profiles in both omics for certain genes. One of the remarkable findings of this heatmap is that the expression of *sarX* and *pur* genes was consistent between transcriptomics and proteomics. The Δ*isaA* mutant downregulated expression of *sarX* and upregulated purine biosynthesis genes *purA*, *purB*, *purC*, *purD*, *purF*, *purH*, *purK*, *purL*, *purM*, *purN* and *purQ*. A study has shown that the *sarX* gene plays an important role in biofilm production because it can activate the *ica* genes, which are associated with biofilm formation [67]. Thus, downregulation of *sarX* in the *isaA* mutant may be another factor related to the significant reduction in biofilm production. A previous study also demonstrated that purine biosynthesis genes might be involved in biofilm formation [68]. They found that the disruption of some *pur* genes resulted in significantly impaired *S. aureus* biofilm formation. However, a clear mechanism regarding the upregulated *pur* genes—which in turn deactivates the other genes, leading to biofilm reduction—is still unknown.

Taken together, our results provide evidence that *isaA* is a gene that affects biofilm formation and the expression of other virulence factors in the biological process of *S. aureus*, especially leukocidins and hemolysins.

## Figures and Tables

**Figure 1 microorganisms-10-01119-f001:**
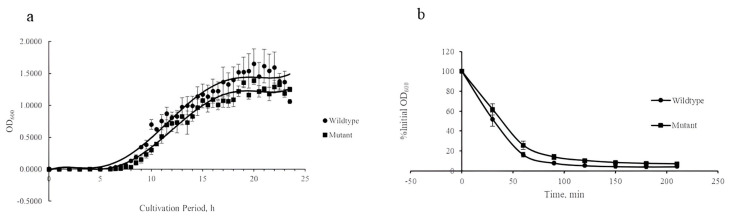
The influence of *isaA* on the growth patterns and autolysis profiles of *S. aureus*. (**a**) The Δ*isaA* mutant and the parental strain SH1000 grown in RPMI medium with similar growth curves. (**b**) Autolysis curve for both bacterial strains, with autolysis activity expressed as percentage of the respective initial OD_600_.

**Figure 2 microorganisms-10-01119-f002:**
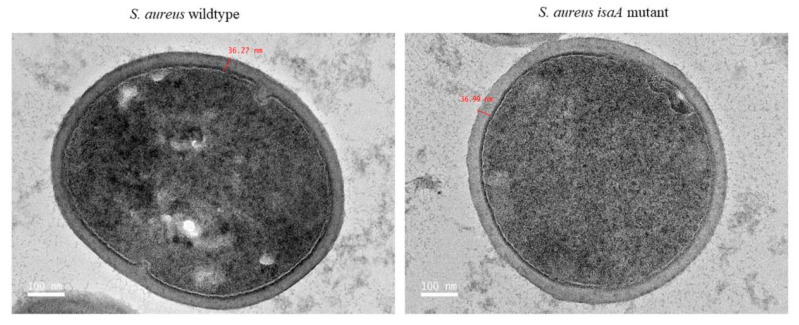
Transmission electron microscopy (TEM) images of *Staphylococcus aureus*. Both wildtype (*S. aureus* SH1000) and *isaA* mutant (*S. aureus* MS001) have intact cell walls. Scale bars: 100 nm.

**Figure 3 microorganisms-10-01119-f003:**
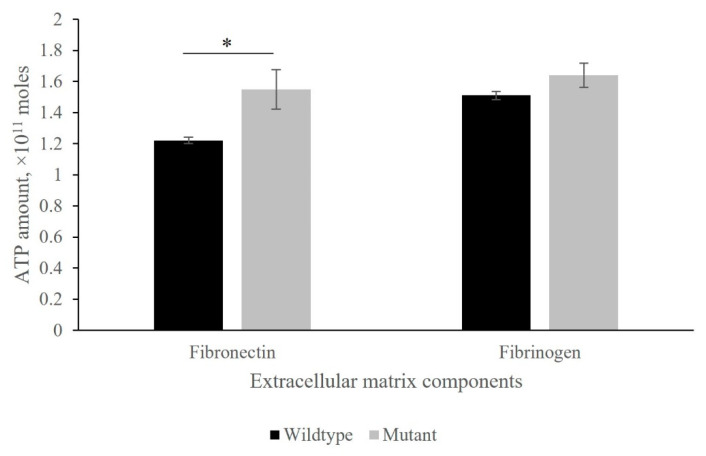
*IsaA* gene disruption affects fibronectin/fibrinogen binding. Adhesion of *S. aureus* Δ*isaA* mutant and its parental strain SH1000 were examined for the immobilised extracellular matrix proteins fibronectin and fibrinogen. This was quantified as ATP amounts in a luminescence microplate reader. High levels of ATP were detected in the *isaA* mutant, with the range of 1.3 × 10^11^ to 2.0 × 10^11^ mole for fibronectin and fibrinogen. * *p* ≤ 0.05.

**Figure 4 microorganisms-10-01119-f004:**
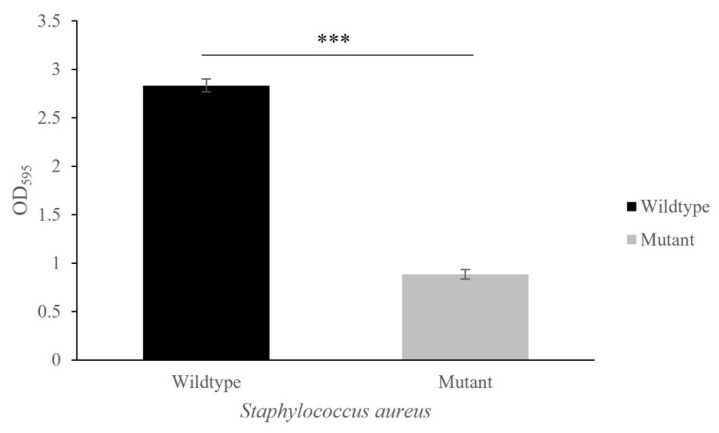
Decreased biofilm formation in *isaA* mutant compared to wildtype. Quantification of biofilm formation was based on the crystal violet retained by bacterial biofilm. *** *p* ≤ 0.001.

**Figure 5 microorganisms-10-01119-f005:**
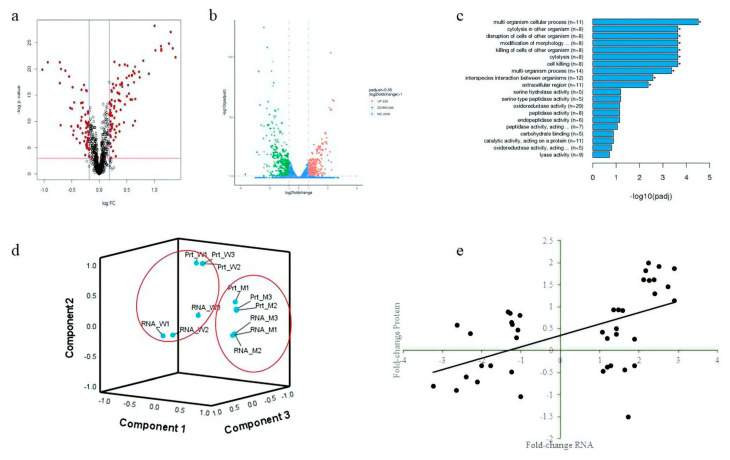
Proteomics and transcriptomics of the *S. aureus* wildtype and *isaA* mutant. (**a**) Volcano plot of differentially expressed proteins (DEPs). The red dots are proteins with significant differences compared to wildtype. (**b**) Volcano plot of differentially expressed genes (DEGs). The pink dots represent the significantly upregulated genes, and the green dots indicate genes that have been significantly downregulated in that Δ*isaA* mutant. (**c**) Gene Ontology (GO) enrichment analysis by clusterProfiler for those differentially expressed genes. (**d**) Principal component analysis (PCA) of significant gene and protein expression data. Wildtype samples are shown within the closed circle on the left, while *isaA* mutant samples are represented by the dots within the closed circle on right. (**e**) Correlation between DEPs and DEGs.

**Figure 6 microorganisms-10-01119-f006:**
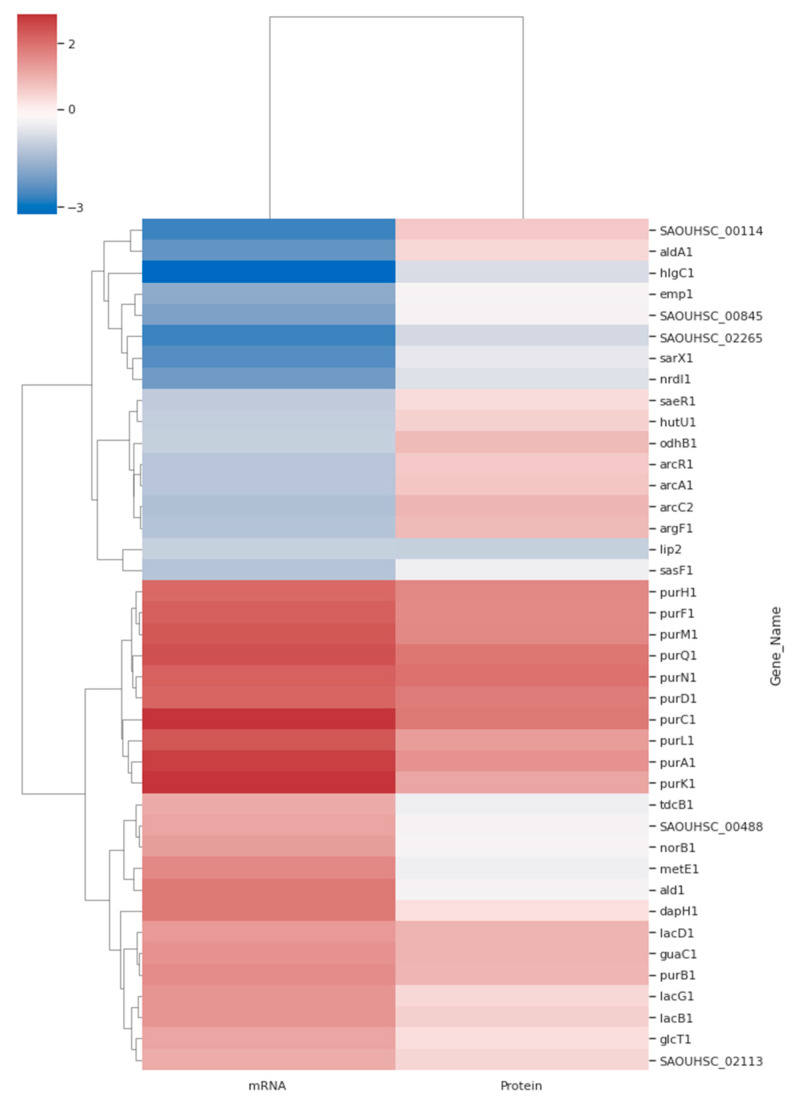
Heatmap of significance of differentially expressed genes and proteins. Forty genes according to matching between proteomic and transcriptomic datasets were used to generate the heatmap. All of these data were log_2_ transformed, and red to blue colour represents high to low expression.

## Data Availability

Not applicable.

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
