# Peer review of "Impact of IsaA Gene Disruption: Decreasing Staphylococcal Biofilm and Alteration of Transcriptomic and Proteomic Profiles"

_microorganisms, 2022, doi:10.3390/microorganisms10061119_

Round 1

Author Response

Dear Prof.,

Please see the attachment. Thanks

Regards,

LYK

Reviewer 2 Report

The research is described clearly and completely. The experimentation starts from a well-posed hypothesis and from an excellent experimental design. The results are very relevant for both research and food safety applications.Very good paper! I congratulate the authors.

 I note only small inconsistencies in the text, which I bring to the attention of the authors.

line 55: MRSA - is bot a single-drug resistant strains as in fact, MRSA is a multiple drug resistance to all beta-lactam antibiotics.

line 94: please add Prof Foster affiliation in parenthesis

line 354: isaA - please use italics 

Author Response

Dear Prof.,

Good day. Please see the attachment. Thanks

Regards,

LYK

Round 2

Reviewer 1 Report

The paper has been improved and can be published. When suggesting update of the reference list, I meant not technical corrections, but inclusion of some recent titles on relevant topics; this could further improve your discussion no matter that it is not a "must".